# Perspectives of healthcare and social support sector policymakers on potential solutions to mitigate financial impact among people with TB in Mozambique: a qualitative study

Pedroso Nhassengo  ,[1,2,3] Clara Yoshino,[1,3] Américo Zandamela  ,[2] Verónica De Carmo,[2] Bo Burström,[1] Celso Khosa,[2] Tom Wingfield,[1,3,4] Knut Lönnroth,[1,3] Salla Atkins[1,5]

**Correspondence to**
Dr Pedroso Nhassengo;
pedroso.nhassengo@ins.gov.mz

## ABSTRACT

**Objective** People with tuberculosis (TB) and their households face severe socioeconomic consequences, which will only be mitigated by intersectoral collaboration, especially between the health and social sectors. Evidence suggests that key factors for successful collaboration include shared goals, trust, commitment, resource allocation, efficient processes and effective communication and motivation among collaborating parties. This study aimed to understand healthcare and social support sector policymakers' perspectives on potential solutions to mitigate financial impact among people with TB and their households in Mozambique.

**Design** Qualitative study with primary data collection through one-to-one in-depth interviews.

**Setting** Gaza and Inhambane provinces, Mozambique.

**Participants** Policymakers in the health and social support sector.

**Results** A total of 27 participants were purposefully sampled. Participants were asked about their perspectives on TB-related financial impact and potential solutions to mitigate such impact. Participants reported that people with TB are not explicitly included in existing social support policies because TB per se is not part of the eligibility criteria. People with TB and underweight or HIV were enrolled in social support schemes providing food or cash. Two themes were generated from the analysis: (1) Policymakers suggested several mitigation solutions, including food and monetary support, but perceived that their implementation would be limited by lack of resources; and (2) lack of shared views or processes related to intersectoral collaboration between health and social support sector hinders design and implementation of social support for people with TB.

**Conclusion** Despite health and social sector policymakers reporting a willingness for intersectoral collaboration to mitigate TB-related financial impact, current approaches were perceived to be unilateral. Collaboration between health and social support sectors should focus on improving existing social support programmes.

## STRENGTHS AND LIMITATIONS OF THIS STUDY

⇒ In-depth interviews conducted by the first author, who had contextual experience and understanding.
⇒ There was a good representation of participants from both provinces, sectors, gender and age.
⇒ Data collection was conducted in the native language of the first author, ensuring comprehensive understanding and quality control.
⇒ Adaptation of data collection tools based on principles of adaptability, flexibility and pragmatism.
⇒ This manuscript did not capture the perspectives of people with tuberculosis.

## INTRODUCTION

Tuberculosis (TB) is one of the leading causes of death due to infection, killing 1.6 million people globally.[1] Until the coronavirus (COVID-19) outbreak, TB was the leading cause of death from a single infectious agent, ranking above HIV/AIDS. Mozambique, which reported 98 000 TB cases in 2021, is on the WHO list of high-TB burden countries that collectively contribute 86% of the global TB burden.[1] The Mozambican health system faces several challenges in addressing this burden including low health provision coverage, low rates of case detection and lack of laboratories for drug susceptibility testing.[1 2] The WHO 2015 'End TB Strategy' aimed for zero TB-affected households to be facing catastrophic costs (defined as total cost exceeding 20% of annual household income) by the year 2020[3 4] and only 52% of TB-affected households were protected by 2021.[1] This focus on the importance of social

support and protecting vulnerable populations was also reinforced in the Sustainable Development Goal 3.3[5] and the United Nations High-Level Meeting on TB.[5 6] In many settings, people with TB incur costs equivalent to more than half of their household's annual income due to TB illness and care seeking.[7 8]

A meta-analysis showed that the proportion of costs among people with TB varies according to the type of TB and co-infection with HIV (TB/HIV). This study further showed that the proportion of people facing catastrophic costs at a cut-off point of 20% of their annual income before TB was 43% among people with susceptible TB and 80% among people with drug resistant TB (DR-TB).[9] Financial impact due to TB and DR-TB has been found to be independently associated with adverse TB treatment outcomes (death during treatment, loss to follow-up, treatment failure or recurrent TB disease within 30 months of starting TB treatment).[10 11] People from low socioeconomic strata or those with DR-TB are at a particularly high risk of severe financial impact.[8 10] Therefore, there is a growing consensus that progress towards the 'End TB Strategy' targets will require not only investment in strengthening biomedical responses through improving access and effectiveness of healthcare delivery[12] but also actions on the socioeconomic determinants and

consequences of TB, as well as measures to improve quality of life and well-being after TB-episode,[3 4 10 13] especially in low- and middle-income countries. The holistic impact of TB on well-being, finances and quality of life requires the simultaneous provision of biomedical and socioeconomic approaches to control TB, which in turn requires intersectoral collaboration especially between healthcare and social sectors.[14–16] However, despite a high proportion of people with TB and their households being economically vulnerable and likely to benefit significantly from social support, they are rarely covered by social support policies.

In Mozambique as in many other settings, TB disease is not a formal criterion for social support, despite its severe consequences for people and families affected by it.[15] To begin developing potential for such support requires collaboration between different sectors. To date, however, there is little knowledge about how a collaboration or pathway between the healthcare and social support sector would, or could, work in practice in Mozambique. Our study therefore focuses on understanding the current landscape of collaboration, and the potential for future collaborations. To the extent of our knowledge, the financial impact faced by people with TB and their households and the role of intersectoral collaborations to address such impact has not previously been studied in Mozambique.

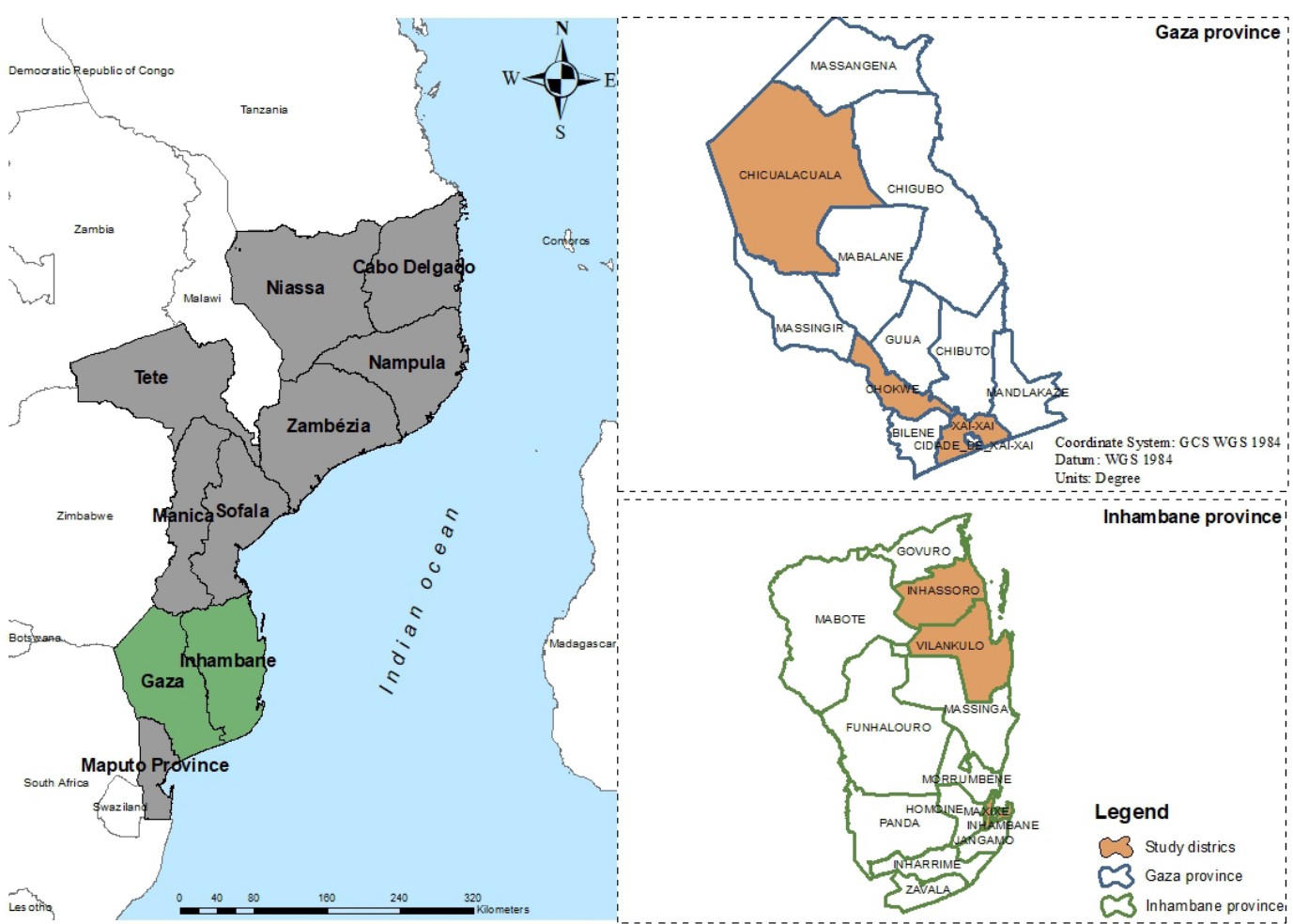

**Figure 1** Map showing the study provinces and districts.

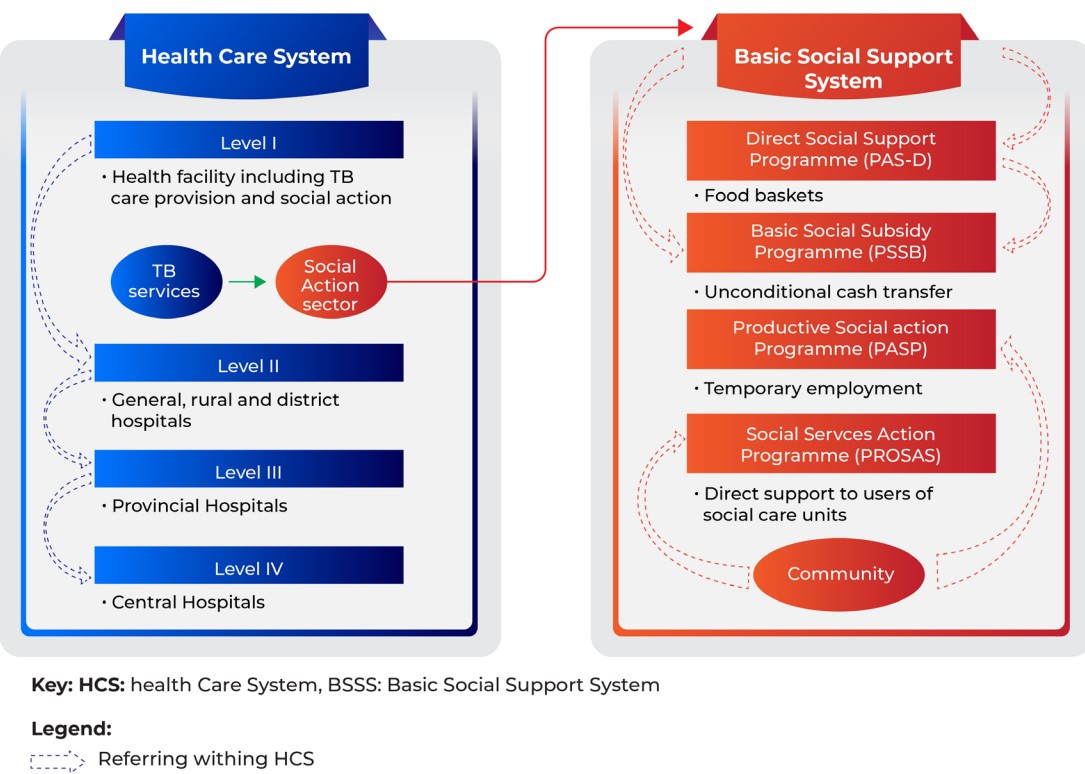

**Key: HCS:** health Care System, BSSS: Basic Social Support System

**Legend:**

- - - - ▷  Referring withing HCS

⟶  Referring from TB Services
   to Social Action Services

⟶  Referring from HCS to BSSS

- - - - ▷  Referring within BSSS

**Figure 2** Current structure of Healthcare and Basic Social Support System. The black arrow represents the referral from the Social Action Sector, a representation of the National Institute of Social Action (INAS) within the health facility with the responsibility of preparing the documents for support requests, to the Basic Social Support System at INAS.

We aimed to understand healthcare and social support sector policymakers' perspectives on potential solutions to mitigate financial impact among people with TB and their households in Mozambique, a low-income and high TB burden country.

## METHODS

### Conceptual framework

The theory underlying this study is that although intersectoral collaboration is complex and sometimes hard to achieve, it will be one of the fundamental issues to consider and address in order to mitigate financial impact of TB-affected households. Common goals, mutual trust and commitment to share the risks, allocation of time and resources are unquestionable preconditions for a sustainable collaboration. Other dimensions to be accounted for include the scope and type of collaboration, efficiency of the collaboration processes and management of the activities and interaction between collaborating parties, which encompasses communication and motivation to collaborate.[16–18]

### Study design

This was a qualitative phenomenological study with primary data collection using one-to-one in-depth interviews with policymakers either in healthcare or in social support sectors. This study was part of the CHEST project (Coordination of HEalth and Social care for TB patients in Mozambique: Policy dialogue and situation analysis) implemented in Mozambique, whose main objective was to estimate the prevalence of people incurring catastrophic costs due to TB and develop a bidirectional referral system between health and social services to increase access of grants by people with TB.

### Involvement of patients and general public

Patients or the public were not involved in the design, conduct, reporting or dissemination plans of our research.

### Setting

Mozambique is a sub-Saharan African country with an estimated population of 31 million people,[19] a literacy rate of 61% and 63.7% living on less than $1.90 a day.[20] Study participants were recruited in seven districts of Inhambane and Gaza provinces in southern Mozambique (figure 1). These provinces had 28 districts that jointly contributed to 12.8% of the country's TB notifications in 2021.[21] Since all 28 districts met the study site inclusion criteria, we randomly selected 2 districts in each province considering the type of districts and geographical representation and, purposely included the provincial

**Table 1** Participants' demographic characteristics

| Demographic characteristics | Gaza n=13 | Inhambane n=14 | Total n=27 |
|---|---|---|---|
| Median age (IQR) | 33 (32–45) | 39 (33–48) | 37 (32–47) |
| Gender n (%) | | | |
| Male | 7 (53.9) | 8 (57.1) | 15 (55.6) |
| Female | 6 (46.2) | 6 (42.7) | 12 (44.4) |
| Work position n (%) | | | |
| Healthcare policymakers | 8 (61.5) | 8 (57.1) | 16 (59.3) |
| District/clinical directors | 3 (23.1) | 2 (14.3) | 5 (18.5) |
| Public health/tuberculosis focal points | 5 (38.5) | 6 (42.9) | 11 (40.7) |
| Social support policymakers | 5 (38.5) | 6 (42.9) | 11 (40.8) |
| Programme managers | 0 (0) | 2 (14.3) | 2 (7.4) |
| Social support focal points | 4 (30.8) | 3 (21.4) | 7 (25.9) |
| District directors | 1 (7.7) | 1 (7.1) | 2 (7.4) |
| Median time in work position in years (IQR) | 4 (3–5) | 3.5 (2–4) | 4 (2–5) |
| Education n (%) | | | |
| Less than secondary school | 0 (0) | 0 (0) | 0 (0) |
| Secondary school | 5 (38.5) | 3 (21.4) | 8 (29.6) |
| University | 8 (61.5) | 11 (78.6) | 19 (70.4) |

representations of both sectors as they present distinct socioeconomic landscapes and Vilanculos as it shared social sector representation with Inhassoro (a selected district). All other selected districts had both health and social sector representation.

## Structure of healthcare and social protection system

The current Mozambican public healthcare services encompass four levels (figure 2). The lower level provides primary healthcare including free-of-charge TB services (diagnosis and treatment)[2] and initial assessment for social support through the Social Action Sector. The Social Support services are delivered by the National Institute of Social Action (INAS) following four operational programmes namely: 1) Basic Social Subsidy Programme (PSSB) providing regular unconditional cash transfer to vulnerable people (mainly people older than 60 and people with disabilities); 2) Direct Social Support Programme (PAS-D) providing food baskets to vulnerable people and households including people with chronic illnesses; 3) Productive Social Action Programme providing temporary employment in urban and rural areas and (4) Social Services Action Programme providing direct support to users of social care units that provides daily meals to elderly people, homeless and orphan in specific places.[22 23] There is no social support scheme available specifically targeted towards TB-affected people and their households. Both healthcare and social support sectors have paper-based registration.

## Study participants and data collection

The participants were purposefully recruited in the selected districts between October 2021 and February 2022 using snowballing and maximum variation sampling.[24] The research team compiled a list of work positions of interest in both sectors. Pre-recruitment was conducted either by email or phone. A total of 34 people were invited to participate, however, 7 were not able to participate due to personal reasons and 2 of those were replaced by an equivalent respondent. Eligibility criteria were (1) age 18 years or older, (2) with at least 1 year of active work in the current position and (3) capable of providing informed consent. Informed consent and data were collected face-to-face by the first author (PN). Interviews were conducted in a quiet and private place at the participant's offices and lasted between 30 and 90 min. We stopped data collection with 27 participants when we reached saturation.[25] Interviews were conducted following a semi-structured topic guide that covered three major sections: (1) perceptions of financial impact faced by people with TB; (2) potential solutions to mitigate the financial impact; and (3) barriers and facilitators for intersectoral collaboration (online supplemental file 1). Interviews were audio-recorded, transcribed *verbatim* and translated from Portuguese into English. Transcriptions were controlled for quality against the recordings by research assistants and against translations by PN. Non-verbal communication and other important information were captured as field notes during each interview.

Of a total of 27 participants enrolled, 14 (52%) were from Gaza and 13 (48%) were from Inhambane province. The overall median age was 37 (IQR: 32–47) years old. Most respondents were men (56%) and the median time at the working position was 4 years (IQR: 3–5). About

**Table 2**  Example of coding

| Theme | Policymakers suggested several mitigation solutions, including food and monetary support, but perceived that their implementation would be limited by lack of resources | | | Lack of shared views or processes related to intersectoral collaboration between health and social protection sector hinders design and implementation of social support for people with tuberculosis | | |
|---|---|---|---|---|---|---|
| Category | Potential solutions for addressing costs | | | Existing collaboration with other sectors | | |
| Codes | Food, cash and transport | Mobile brigades and expansion of quality healthcare system | Better housing and psychological support | Mechanisms of collaboration | Involved stakeholders | Type of collaborating institutions |

three quarters of the participants had obtained a university degree (table 1).

## Data analysis

Qualitative content analysis was adopted as proposed by Graneheim and Lundman.[26] Data were deductive and inductively coded by PN and CY using Dedoose software and analysed by PN, CY and SA. Three interviews were double coded to verify the similarity of the encodings. For independent coding, the researchers had regular discussions about the understanding of the codes. The deductive coding was based on the topic guide and preliminary analysis of the interviews. This was followed by inductive coding with the inclusion of codes that emerged from the data during the deductive coding. The interview text was sorted into four content areas: (1) incurred cost and mitigation; (2) types of support for people with TB; (3) benefits of supporting people with TB and (4) intersectoral collaboration. Texts corresponding to these areas were extracted from the interview transcriptions and brought together into one text, which formed the unit of analysis. The text was divided into meaning units that were condensed. The condensed meaning units were, abstracted and labelled with a code. Through a manifest analysis, relevant codes to the research questions were rearranged and grouped into categories (table 2). The last stage consisted of interpreting the data through a latent analysis and developing the themes. Full coding process is presented in the online supplemental file 2.

## Researcher characteristics and reflexivity

The interviews were conducted by PN. He is a medical doctor and PhD candidate with 7 years of experience in qualitative research. PN's research is based in Mozambique, and he is familiar with the health and social services system. Coding was conducted by PN and CY, who are both native Portuguese speakers and fluent English speakers. She is a social scientist and public policy researcher. CY and SA have experience in qualitative and international social protection systems.

PN, being affiliated with a central public institution, recognised that conducting research at the provincial level could influence participants to provide favourable opinions. Therefore, PN invested significant effort in explaining study procedures, obtaining informed consent and addressing participants' concerns. The interview

began only after establishing rapport and trust between PN and the participants.

CY's experiences of similar health and social services from Brazil added value to the analysis. She immersed herself in the data and got a thorough understanding of its content and context. The team's experience on TB and social protection in Africa and globally improved the understanding of the research findings.

## Ethical considerations

All participants were provided detailed information about the study and provided a written informed consent before the interview. The study was performed in accordance with the study protocol, the Declaration of Helsinki (last update: October 2013), was approved by the Institutional Ethics Committee of the *Instituto Nacional de Saúde* in Maputo, Mozambique (Ref. no. 001/CIBS-INS/2020) and by the Swedish Ethical Review Authority in Stockholm, Sweden (Ref. no. 2022-03297-01). Interviews were anonymised to protect participant's identity.

## RESULTS

### Perceptions of financial impact faced by people with TB and its consequences

Healthcare and social sector policymakers had different understanding of the need and provision of support for people with TB. Social sector's policymakers had little knowledge of the treatment and care pathways of people with TB and thus their need for support. On the other hand, healthcare policymakers were aware of the impact of TB on people's lives and the socioeconomic implications of treatment. They reported that TB-related costs were high for people with TB and their households, and this was worse when the Primary Income Earner (defined as the person with the largest economic contribution to secure the family's basic needs) was affected or the person had DR-TB. Incurred costs due to transportation, food and income loss seemed to be the major causes of negative financial impact of TB.

**Theme 1: policymakers suggested several mitigation solutions, including food and monetary support, but perceived that their implementation would be limited by lack of resources**

*Current availability and source of social support provision*

The current availability of support for people with TB was seen as inadequate by both healthcare and social sector policymakers. While institutional actions in the healthcare sector included active tracing by community workers and decentralisation of TB services to remote areas, the social protection sector provided food and monetary support through PAS-D and PSSB. Beyond TB services-related measures, the healthcare sector could not recount other support. Both sectors mentioned the potential benefit of psychological support to people with TB and their families focusing on the disease, treatment of TB and enrichment of food through practical mentoring about their diet. This was perceived as having positive effects including decreasing the need for support and demand for supplements at health facilities.

*Community centred strategies and expansion of TB services*

Healthcare policymakers described extensively the community-based approaches that they perceived to reduce direct and indirect out-of-pocket medical costs. They mentioned, for example, brigades, which are a group of healthcare workers whose responsibilities can include community contact tracing, community-based directly observed therapy, TB screening and sample collection, home visits and follow-up of people under treatment. Community-based strategies were seen as key to people with TB mainly due to reduction of travelling costs.

> One of our ways to reduce the cost [of TB treatment] is the community DOT where we have our activists there. […] this way we reduce the cost of [people with TB] who have to come to the health facility twice a month. (Male, healthcare policymaker, 3 years in position)

Expansion of healthcare services to more remote areas was considered by healthcare policymakers as a suitable strategy to reduce transportation cost and waiting time at the health facilities. They reported the expansion of TB treatment sites having as example the antiretroviral therapy (ART) services, which was expanded to all health facilities and resulted in perceived reduction of transportation cost. They also suggested that counsellors (staff dedicated to counselling at health facilities or at communities) and health technicians (staff with basic medical training and abilities to diagnose and prescribe medications for the most common diseases) should work together as community health workers. The current approach for community work was seen as inappropriate as the healthcare sector was mainly represented by counsellors who lack clinical knowledge. On the other hand, health technicians or community health workers were perceived to be cheaper than medical doctors.

*Food provision and monetary support for people with TB*

According to both healthcare and social sector policymakers, the existing policy on provision of social support for people with TB was conditional on having other conditions such as HIV and underweight and they were eligible to receive food supplements, as part of a collaboration with the nutrition programme.

From the social sector's side, people on ART or chronically ill (illnesses requiring long-term care and support) received a basic food basket containing nutritional supplementation through the PAS-D. The programme initially covers 6 months of continuous support after which recipients are reassessed regarding the need for continued support. If so, the support could be extended for additional 6 months or the recipient would be redirected to the PSSB, which offers lifelong monetary support. However, having TB was not per se an eligibility criterion for enrolling in this programme.

One social sector policymaker (female, social sector policymaker, 3 years in position) reported that people with chronic illnesses used to receive food baskets, but due to lack of funds, they started to receive cash, as they have a lower cost compared with PAS-D food baskets. They also mentioned that there is a direct cash support for transport. However, people with TB face difficulties in receiving such support, as it requires them to provide proof of transport payment, which is seldom available on public transportation.

*Barriers and facilitators of addressing TB-related costs*

Participants reported different strategies undertaken by both institutions and households to mitigate TB-related financial impact. Healthcare policymakers reported strategies such as obtaining additional income by selling livestock and spouses whose husband with TB was primary income earner taking over financial responsibilities.

Cost mitigation activities conducted by both policymakers included institutionalised and non-institutionalised actions. Institutionalised actions were defined as activities implemented according to the operational procedures and non-institutionalised actions were defined as activities or actions implemented by policymakers' instant initiative. The latter resulted from participants being sensitised by the situation of the person with TB and helping them using their own resources. These consisted of different levels of involvement, from healthcare policymakers advising people with TB on possible ways to obtain some income to mobilising the community to donate food or money. Social sectors' policymakers also reported trying to enrol people with TB in any social protection programme.

Strategies to mitigate TB-related financial impact were reported by healthcare policymakers as still insufficient, as people with TB continue to incur high costs and therefore risk to abandon the treatment.

*Potential solutions for addressing TB-related costs*

Both healthcare and social sector policymakers thought that food was the most important social support option for people with TB, because the need for food cannot be compensated through other means and many people with TB were food insecure. Lack of transport, for example, could be replaced by walking in the absence of cash; but food is a staple that is always needed by people with TB. Healthcare policymakers also reported that food was the most common form of support that people with TB asked for. However, food management was considered complicated, as it requires more complex logistics (purchase, storage and distribution).

> I would prioritise food because we end up having other alternatives for transport but for food, we don't have any mean that helps us to minimise the situation of [people with TB]. (Male, healthcare policymaker, 6 years in position)

According to policymakers, the lack of food could be countered either by direct provision of food or by provision of cash. However, healthcare policymakers were reluctant to give cash due to the perceived risk of misuse, such as alcohol consumption. Despite this, social sector reported that cash provision would be practically and logistically easier and that most people would adequately use the cash, as they were instructed during psychological support. Both healthcare and social sector policymakers seemed to believe that addressing only other financial burdens, such as transportation cost, is not enough if food was not provided.

> [...] money, as such, I don't think so. Money creates problems ahead, that's why I am not suggesting money support. A basic food basket is better because when we give money to certain people who are undergoing treatment when they start to improve instead of buying food, they deviate the money. (Male, healthcare policymaker, 3 years in position)

### Theme 2: lack of shared views or processes related to intersectoral collaboration between health and social protection sector hinders design and implementation of social support for people with TB

*Barriers and facilitators to collaboration between healthcare and social sectors*

Participants perceived involvement of key stakeholders as an important step towards appropriate cost-mitigation strategies. The Ministry of Health, through its provincial and district representations and the Ministry of Gender, Children and Social Action, through INAS and its provincial and district representations are considered the most important stakeholders. Other governmental stakeholders played complementary roles. The community and churches providing informal social support were also considered important players. Participants also suggested that all stakeholders working in similar areas or in philanthropic projects should be engaged to strengthen the collaborations.

Support for people with disabilities or chronic illness was predominant in the collaboration between the healthcare and social institutions. Both healthcare and social sector participants mentioned that there was an ongoing intersectoral collaboration for supporting people with chronic illnesses. A substantial barrier was seen as the non-inclusion of TB disease in Mozambican social support policies which prevented direct provision of social support to people with TB. An alternative measure was registering people with TB according to other chronic illness or disability that they might have such as chronic lung disease or malnutrition. Overall, healthcare policymakers seemed dissatisfied with the current collaboration. Waiting time for support to be provided, poor communication, perceived lack of resources and cumbersome vulnerability assessment were reported as being the source of dissatisfaction. They described waiting time for support provision as long and inadequate for people with TB and opined that often the social sector counterpart delayed the feedback of support applications.

> The most obvious barrier is the delay in response [...] because the feedback of the applications take time. At some point, we get the response while the patient has already finished the treatment or maybe others have already defaulted or died. (Female, healthcare policymaker, 33 years in position)

Participants from the healthcare sector seemed to not understand how the social sector proceeds for support allocation. Information about the correct steps as well as the eligibility for support provision seemed to be scarce. On the other hand, the social sector assists very few people with chronic illness (including TB), compared with elderly people and children. This may be explained by limited dissemination of the programmes, lack of understanding and low referral rates by the health sector. Additionally, both participants reported resource limitations to cover new applications. Allocated budget and goals were predefined at central level and new beneficiaries entered the programmes only when current beneficiaries were discharged from it resulting in long waiting list.

On top of this, the lifelong support provided to beneficiaries with chronic illnesses was perceived to be a small amount varying according to the number of household members (from $8 to $16 per month). Almost all participants agreed that the amount was insufficient to handle basic needs even for a single-person household.

Participants from social sector stressed that the sector, understandably, lacks clinical knowledge or competence. Participants reported that a health technician or a psychologist allocated to their offices would improve the identification of sick beneficiaries at the community level and promptly refer them to the healthcare sector for further management as appropriate.

Intersectoral communication was raised as an area for improvement by healthcare sector but not social sector participants. One healthcare participant suggested that the intersectoral communication could be more informal

and formalities could be left to a later stage and that high-level intersectoral policymaking meetings and discussions could be of benefit to define common goals and consider how best intersectoral collaboration could successfully be implemented and achieved.

### Mechanisms of collaboration between healthcare and social sectors

The roles of the two major players in supporting people with TB seemed clear to all participants. The health sector's responsibility was to diagnose the TB disease, check clinical eligibility and pre-assessment of vulnerability, which enabled them to refer these people to the Social Action Sector. On the other hand, the Social Action Sector should do a more complete and detailed vulnerability assessment and submit the application to the social sector at INAS. The expectations were that INAS would provide approval for enrolment and support provision if the budget was available. Another direction of collaboration was related to how INAS' beneficiaries reach the health sector when they become sick (eg, people with presumptive TB). This was unclear and may not be in place as INAS' lack of competence to assess health status limited them to counselling and advising people to go to the health facilities for better assessment and diagnosis.

## DISCUSSION

This study revealed healthcare and social sector policymakers' perceptions of the financial impact and challenges faced by people with TB in Mozambique as well as current and potential joint mitigation strategies between the two sectors. Since TB is not considered an eligibility criterion in current Mozambican social support policies, people with TB and their households are not covered by adequate and equitable social support. Our findings provide valuable information about both the current provision of social support and gaps in support for people with TB, suggested solutions for support provision during treatment phase and dynamics of intersectoral collaboration between healthcare and social support sector.

People from low socioeconomic strata are disproportionately affected by TB, and our interviewees were clear, at least in the healthcare sector, that social determinants are the drivers of TB in Mozambique. Similar to other studies, there were vulnerabilities among people with TB that included poverty, low income, food insecurity, undernutrition[27] and risk of income loss. This was due to their livelihoods being based predominantly on farming, self-employment or informal work, with many unable to benefit from stable labour activities[28] and health and sickness insurance coverage. This financial situation meant an inability to afford the costs associated with TB. Studies in Africa and Asia also reported similar income and job-related problems.[29] Distance to health facilities and its related transportation costs were perceived as negatively impacting health seeking behaviour, leading to delays in presenting at health facilities, worsening of health condition and risk of spreading the disease to cohabiting people.[7] Health system-related delays in diagnosis also contribute to increasing the financial impact due to costs with transportation and out-of-pocket payments of medication taken during waiting time. We also found that healthcare policymakers thought that the TB-related costs and consequences were exacerbated by rigid DOT routines. Most people with TB struggle to have enough food and money for daily transportation to DOT sites. Distance and transportation cost to the DOT site, costs of food, out-of-pocket medication and income loss were the leading causes of high costs during treatment and lack of transportation money during TB treatment was associated with missing DOT visits and TB doses and lack of food with treatment failures.[8]

Although lack of food and money negatively impacted the TB treatment, food provision was the most important due to its impact on treatment adherence and medication intake. Similar studies showed that food support was the most important for more easily averting the challenges faced by people with TB. However, our findings indicate that food provision requires a more complex logistical apparatus for purchasing, storage and distribution to beneficiaries making the cash preferred over food support and cash-based schemes are less costly than food-based interventions.[30] The impact of existing cash support schemes was reported to be limited by low cash amounts provided. A systematic review by Todd *et al* discussed the consensus that that the cash transferred value should be large enough to mitigate the TB-related financial impact and motivate households to remain in care, while being too small to potentially act as a coercion.[31] Psychological support currently provided by healthcare and social support sectors to people with TB and their household members was considered indispensable, but insufficient by itself. A study by Ahuja *et al* in western India[32] showed that family support to people with TB improved their ability to remain in care. However, South African evidence suggests that while family can be an important support structure, as their resources are limited, people with TB are left without important sources of support.

Ultimately, existing collaborations between the two key players could improve referral and support for people with TB and their households. However, lack of local evidence of disease-related financial and social impact might hamper better understanding of the need for urgent intervention and collaboration. Alignment of aims and goals between the healthcare and social support sector was likely to have been influenced by lack of official collaboration establishment, definition of roles of the parties, specific action plan to be carried out, benefits and necessary resources.[16] Collaboration should be established and regulated at central level ensuring definition of goals and roles of the parties. Local level institutions should adapt the collaboration to fit the available resources and needs of the affected people.

In light of this, locally based research to assess the TB-related financial and social impact is required to

support the inclusion of people with TB in the currently available social support schemes and improve the collaboration and actions at the policymaking level.

## Strengths and limitations

Strengths of this study included good representation of policymakers both in terms of provinces, sectors, gender and age, high quality and depth of interviews and data collection conducted by PN. This allowed a comprehensive understanding of the responses and quality control of the transcripts in Portuguese and its translation to English. Additionally, data collection tools were adapted during the process following the principles of adaptability, flexibility and pragmatism. An important limitation of this study was the timing of the data collection, which coincided with the period when state servants usually go for annual leave. Because of this, some potential study participants could not be reached. In this situation, the interview was made with the person responding to the position at the time of data collection. This manuscript did not capture the perspectives of people with TB, an important information to triangulate our data. Despite this, our findings are consistent with existing literature discussing solutions of TB-related costs.

## CONCLUSIONS

TB disease is still perceived as a disease of poverty, driven by social determinants and with severe socioeconomic consequences. Our findings from key health and social sector policymakers suggest that the two sectors are willing to collaborate beyond the current modest and unilateral efforts to mitigate the financial hardship of TB. Collaboration across sectors should be strengthened and eligibility criteria for social support should be revised to explicitly include people with TB to enable mitigation of costs of care and compensation of foregone income. Lack of financial and human resources hampers implementation of several identified strategies that could potentially avert the TB-related cost faced by people with TB and their households. Further research is needed to explore reasons of non-inclusion of patients with TB as eligible beneficiaries, as well as barriers for broader policy changes for social protection inclusion of people with TB.

**Author affiliations**
[1]WHO Collaborating Centre in Tuberculosis and Social Medicine, Department of Global Public Health, Karolinska Institute, Stockholm, Sweden
[2]Instituto Nacional de Saúde, Marracuene, Mozambique
[3]Health and Social Protection Action Research and Knowledge Sharing Network, Stockholm, Sweden
[4]Department of International Public Health and Clinical Sciences, Liverpool School of Tropical Medicine, Liverpool, UK
[5]New Social Research and Faculty of Social Sciences, Tampere University, Tampere, Finland

**Acknowledgements** The authors express their gratitude to the Ministry of Health and the Ministry of Gender, Children and Social Action from Mozambique and study participants for their participation and support. We also would like to thank the Mozambican National Tuberculosis Programme for its contribution to setting up the study. Lastly, we would like to thank Dr Kerry Viney for her support in designing the study and reviewing the paper.

**Contributors** Fund acquisition and study design: SA, PN, CK, TW, BB and KL. Data collection: PN, CY, AZ, VC, BB, CK, TW, KL and SA. Data analysis: PN, CY, AZ, VC and SA. Writing of the first draft: PN, CY, AZ, VC and SA. Review and editing of the manuscript: PN, CY, AZ, VC, BB, CK, TW, KL and SA. Guarantor: PN. All authors read and approved the final version of the manuscript.

**Funding** This study was funded by Vetenskapsrådet through the Swedish Research Council, VR 2017-05497. TW is supported by grants from the Wellcome Trust, UK (209075/Z/17/Z), the Medical Research Council, Department for International Development and Wellcome Trust (Joint Global Health Trials, MR/V004832/1), and a Dorothy Temple Cross Tuberculosis International Collaboration Grant from the Medical Research Foundation (MRF-131-0006-RG-KHOS-C0942), UK.

**Map disclaimer** The inclusion of any map (including the depiction of any boundaries therein), or of any geographical or locational reference, does not imply the expression of any opinion whatsoever on the part of BMJ concerning the legal status of any country, territory, jurisdiction or area or of its authorities. Any such expression remains solely that of the relevant source and is not endorsed by BMJ. Maps are provided without any warranty of any kind, either express or implied.

**Competing interests** None declared.

**Patient and public involvement** Patients and/or the public were not involved in the design, or conduct, or reporting, or dissemination plans of this research.

**Patient consent for publication** Not applicable.

**Ethics approval** This study was approved by the Institutional Ethics Committee of the Instituto Nacional de Saúde in Maputo, Mozambique (Ref. no. 001/CIBS-INS/2020) and by the Swedish Ethical Review Authority in Stockholm, Sweden (Ref. no. 2022-03297-01). Participants gave informed consent to participate in the study before taking part.

**Provenance and peer review** Not commissioned; externally peer reviewed.

**Data availability statement** No data are available.

**ORCID iDs**
Pedroso Nhassengo http://orcid.org/0000-0002-2519-1419
Américo Zandamela http://orcid.org/0000-0003-0698-6686

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
