## [Reviewer comments · BMJ Open]

ARTICLE DETAILS

TITLE (PROVISIONAL)	Perspectives of healthcare and social support sector policymakers on potential solutions to mitigate financial impact among people with TB in Mozambique: a qualitative study
AUTHORS	Nhassengo, Pedroso; Yoshino, Clara; Zandamela, Américo; De Carmo, Verónica; Burström, Bo; Khosa, Celso; Wingfield, Tom; Lönnroth, Knut; Atkins, Salla

VERSION 1 – REVIEW

REVIEWER	Rupani, Mihir Government Medical College Bhavnagar, Community Medicine
REVIEW RETURNED	17-May-2023

GENERAL COMMENTS	Title and abstract 1. Abstract: a brief on the theoretical framework used for the study should be added.2. Abstract: results should include some solutions to mitigate the impact, which the current results section is missing.3. Abstract: the conclusion should include a sentence informing the national TB program in Mozambique about some policy changes which the program managers and stakeholders should take at the national level. (for e.g. cash transfer scheme for patients with TB) Introduction 1. The global TB report 2023 is released. Authors should cite recent references and update the figures accordingly.2. "In many settings, people with TB incur costs equivalent to more than half of their household's annual income due to TB illness and care seeking" - consider citing this recent SR-MA (https://pubmed.ncbi.nlm.nih.gov/35017604/). Consider citing this for drug-sensitive TB (https://idpjournal.biomedcentral.com/articles/10.1186/s40249-020-00760-w).3. Social support, I believe, is a broader concept. Authors can consider citing examples from other LMICs, wherein, cash transfer schemes have been proven to improve treatment outcomes of patients with TB (https://www.ijhpm.com/article_4192.html), as one form of social support. Since many countries are implementing cash transfer schemes for patients with TB, a paragraph on the same should be added to the discussion section, culminating with some guidance on how Mozambique can consider implementing such a scheme.4. Prevalence of catastrophic costs due to TB in Mozambique have not been cited, probably, there's a lack of evidence. If it is so, it should be highlighted in introduction section. In discussion section, the need to generate evidence from a nation-wide study on the same should be highlighted. Materials and methods 1. Methods section should start with a theoretical framework used in this study.
---

	Results 1. Was there a felt need among the healthcare providers/program managers regarding the need for social protection for patients with TB? Discussion 1. Discussion section is not well-substantiated (for e.g., the last paragraph just before 'strengths and limitations' is just a repeat of the results, which has been already summarized in the first paragraph). I feel this manuscript is an excellent opportunity to highlight different facets affecting TB-related patient costs and ways to address them. 2. A limitation of the study is the lack of the perspectives of patients with TB in Mozambique, which would have provided a good triangulation of the findings. Authors should add this as a limitation of this study. 3. TB comorbidities add to the costs incurred by the patients. Authors should discuss the additional costs added by the comorbidities such as HIV and diabetes, with an overall scenario of the prevalence of these diseases (TB-HIV and TB-diabetes) in Mozambique and other LMICs. 4. Apart from acting as a financial cushion, especially during the initial phase of the treatment, cash transfer schemes have demonstrated to act as a motivation to continue treatment and thereby help in improving treatment adherence (and thereby treatment outcomes). Authors should add this to the discussion section. 5. Evidence has suggested to add nutritional supplements along with the monetary assistance. While discussing this, authors should highlight the future possibility of universal cash transfers which addresses poverty and other social determinants of health, and thereby helping to reduce the burden of TB (https://www.njcmindia.com/index.php/file/article/view/2225, https://journals.plos.org/plosmedicine/article?id=10.1371/journal.pmed.1002418). 6. Authors should also highlight how this collaboration between health and social sector is possible in Mozambique. At what level(s), and how? Conclusion 1. Future research should also demonstrate this collaboration in Mozambique. Additional comments: References need to be corrected (for e.g. the first reference is not correctly cited). References after 36 are not listed. And, the scope for improvement and list of incomplete references goes on.
--	--

REVIEWER	Ajudua, Febisola I. Nelson Mandela Metropolitan University
REVIEW RETURNED	22-May-2023

GENERAL COMMENTS	The study highlights the gap in provision of social support services to patients diagnosed with TB in Mozambique, a high TB burden country. The introduction adequately outlines the social and scientific value for this study. Although the diagram in page 21 outlines the contextual framework for the provision of support services, it is not titled. It is unclear whether this is what obtains currently, or what the author proposes for the provision of social support services to TB patients in need. The author identified policy makers in areas identified to have the highest incidence of TB disease and describes how this is done. The author also demonstrates an understanding of the study method and clearly states this was a series of semi-structured interviews of selected policymakers. The limitation to the study method is described.
---

	The findings are outlined clearly and the author uses adequate statistical instruments in the review of quantitative data. The discussion draws adequate comparison with international studies. It also addresses the main findings highlighting the effect of inadequate social support on health seeking behaviour, how the lack of social support could delay linkage to care and the need for better alignment of goals between the health and social support sectors. Although mentioned, it is not clear how policy makers in social support services can have a better understanding of the import of social support for TB patients. The writing is clear and articulates the important aspects of reporting for a qualitative study. However, a few areas of the writing require attention. In the abstract, page 3, Line 16,17 – The authors need to specify semi-structured or in-depth interviews The references are relevant to the subject but a number of references are inaccurate in presentation provided on the list. Please pay particular attention to the listed references 1, 21, 25 and 35. Diagram in page 21 is not titled, what area in the writing is it relevant for?
--	--

VERSION 1 – AUTHOR RESPONSE

REVIEWER: 1

Dr. Mihir Rupani, Government Medical College Bhavnagar

A. Title and abstract

1. Abstract: a brief on the theoretical framework used for the study should be added.

Thank you for the recommendation, The theoretical framework underlying the study was added on the abstract. The added text reads as follows: “Evidence suggests that key factors for successful collaboration include shared goals, trust, commitment, resource allocation, efficient processes, and effective communication and motivation among collaborating parties”. Please, see lines 5-7 on page number 2.

2. Abstract: results should include some solutions to mitigate the impact, which the current results section is missing.

The potential solutions to mitigate the impact was added to this section.

3. Abstract: the conclusion should include a sentence informing the national TB program in Mozambique about some policy changes which the program managers and stakeholders should take at the national level. (for e.g. cash transfer scheme for patients with TB).

Thank you for your recommendation. We have included a sentence recommending improvement of the currently existing collaborations across sectors aiming to better mitigation of the impact related to TB disease. We understand that unilateral changes by the National TB Program might not be sufficient to tackle the financial hardship without efficient collaboration with the social support sector. We, therefore, reviewed the sentence to better reflect our understanding. The added sentence reads as follows: “Collaboration between health and social support sectors should focus on improving existing social support programmes”. Please, see lines 25-26 on page number 2.

B. Introduction

1. The global TB report 2023 is released. Authors should cite recent references and update the figures accordingly.

Thank you for your recommendation. We did our best to cite the relevant literature existing by the time of the submission to the journal and, we also reviewed the citations to reflect the currently existing literature. The WHO Global TB Report 2022, the most up to date report available, was cited.

2. "In many settings, people with TB incur costs equivalent to more than half of their household's annual income due to TB illness and care seeking" - consider citing this recent SR-MA (<https://pubmed.ncbi.nlm.nih.gov/35017604/>). Consider citing this for drug-sensitive TB (<https://idpjournal.biomedcentral.com/articles/10.1186/s40249-020-00760-w>).

Thank you for your comment and input. We added one of the suggested articles to our reference. A sentence highlighting the findings from this paper was added to the background. The added sentence reads as follows: "A meta-analysis showed that the proportion of costs among people with TB varies according to the type of TB and co-infection with HIV (TB/HIV). This study further showed that the proportion of people facing catastrophic costs at a cutoff point of 20% of their annual income before TB was 43% among people with Susceptible TB and 80% among people with Drug Resistant TB (DR-TB)". Please, see lines 17-21 on page number 3.

3. Social support, I believe, is a broader concept. Authors can consider citing examples from other LMICs, wherein, cash transfer schemes have been proven to improve treatment outcomes of patients with TB (https://www.ijhpm.com/article_4192.html), as one form of social support. Since many countries are implementing cash transfer schemes for patients with TB, a paragraph on the same should be added to the discussion section, culminating with some guidance on how Mozambique can consider implementing such a scheme.

Thank you for your comments. We added a sentence that improves our statement regarding the importance of cash transfer schemes and the limitation that the sectors and the country is currently facing. The added text reads as follows: "In Mozambique as in many other settings, TB disease is not a formal criterion for social support, despite its severe consequences for people and families affected by it". Please, see lines 36-37 on page number 3.

4. Prevalence of catastrophic costs due to TB in Mozambique have not been cited, probably, there's a lack of evidence. If it is so, it should be highlighted in introduction section. In discussion section, the need to generate evidence from a nation-wide study on the same should be highlighted.

Thank you for your comment. To the extent of our knowledge, there is not any available data on catastrophic cost in Mozambique. A sentence stating lack of evidence in Mozambique was added in the introduction and another paragraph stating the need to generate evidence was added in the discussion section. The added sentence reads as follows: "To the extent of our knowledge, financial impact faced by people with TB and their households and the role of intersectoral collaborations to address such impact has not previously been studied in Mozambique". Please, see lines 42-44 on page number 3.

C. Materials and methods

1. Methods section should start with a theoretical framework used in this study.

A sub-section entitled "Conceptual framework" was added to the methods section. The added sub-section reads as follows: "The theory underlying this study is that although intersectoral collaboration is complex and sometimes hard to achieve, it will be one of the fundamental issues to consider and address in order to mitigate financial impact of TB-affected households. Common goals, mutual trust and commitment to share the risks, allocation of time and resources are unquestionable preconditions

for a sustainable collaboration. Other dimensions to be accounted for include the scope and type of collaboration, efficiency of the collaboration processes and management of the activities and interaction between collaborating parties, which encompasses communication and motivation to collaborate". Please, see lines 7-14 on page number 4.

D. Results

1. Was there a felt need among the healthcare providers/program managers regarding the need for social protection for patients with TB?

Thank you for your comment. Both Providers and Program managers had a clear understanding of the need for social protection for people with TB. Participants from health sector considered the current strategies as insufficient as people with TB and their families still face high cost. These findings were discussed under Theme 1.

E. Discussion

1. Discussion section is not well-substantiated (for e.g., the last paragraph just before 'strengths and limitations' is just a repeat of the results, which has been already summarized in the first paragraph). I feel this manuscript is an excellent opportunity to highlight different facets affecting TB-related patient costs and ways to address them.

The discussion section was revised, and new literature were added. Although we reported the TB-related patient cost on this manuscript, its focus was on collaboration between health and social sectors and the perspectives of policymakers.

2. A limitation of the study is the lack of the perspectives of patients with TB in Mozambique, which would have provided a good triangulation of the findings. Authors should add this as a limitation of this study.

This study is part of a larger project called CHEST, which includes qualitative and quantitative studies. One of the qualitative studies covers the perspectives of people with TB. We included this piece of information in the Methods (please see lines 17- 21 on page 4) and addressed it in the Limitations (please, see line 18-21 on page 14).

3. TB comorbidities add to the costs incurred by the patients. Authors should discuss the additional costs added by the comorbidities such as HIV and diabetes, with an overall scenario of the prevalence of these diseases (TB-HIV and TB-diabetes) in Mozambique and other LMICs. We appreciate this comment. This study was focused on collaboration between health and social sectors and the perspectives of policymakers. Since additional costs related to comorbidities from TB were not one of the objectives of the study, we did not include it in this manuscript. However, there is one quantitative study in the CHEST project that is focused on costs. It aims to estimate the prevalence of people incurring catastrophic costs due to TB in the same setting. We addressed the additional costs led by other diseases and TB comorbidities in other manuscript of the CHEST study.

4. Apart from acting as a financial cushion, especially during the initial phase of the treatment, cash transfer schemes have demonstrated to act as a motivation to continue treatment and thereby help in improving treatment adherence (and thereby treatment outcomes). Authors should add this to the discussion section.

We appreciate this comment. We added a sentence about the perceived preference of the cash transfer as it requires less logistical effort, and it was showed by previous studies to be less costly and included the potential effect in improving motivation to remain in care. The added sentence reads as follows: "our findings indicate that food provision requires a more complex logistical apparatus for purchasing, storage, and distribution to beneficiaries making the cash preferred over food support and cash-based schemes are less costly than food-based interventions". Please, see line 25-28 on page

number 13.

5. Evidence has suggested to add nutritional supplements along with the monetary assistance. While discussing this, authors should highlight the future possibility of universal cash transfers which addresses poverty and other social determinants of health, and thereby helping to reduce the burden of TB (<https://www.njcmindia.com/index.php/file/article/view/2225>, <https://journals.plos.org/plosmedicine/article?id=10.1371/journal.pmed.1002418>).

We appreciate your comment. This study was focused on collaboration between health and social sectors and the perspectives of policymakers. Food provision and cash support will be discussed into details in the upcoming CHEST manuscript.

6. Authors should also highlight how this collaboration between health and social sector is possible in Mozambique. At what level(s), and how?

We appreciate your comment. A statement about the collaboration was added in discussion section. The added statement reads as follows: Collaboration should be established and regulated at central level ensuring definition of goals and roles of the parties. Local level institutions should adapt the collaboration to fit the available resources and needs of the affected people. Please, see line 2-4 on page number 14.

F. Conclusion

1. Future research should also demonstrate this collaboration in Mozambique.

A statement about the need for further studies was added in conclusions section. The added statement reads as follows: Further research is needed to explore reasons of non-inclusion of TB patients as eligible beneficiaries, as well as barriers for broader policy changes for social protection inclusion of people with TB. Please, see line 32-34 on page number 14.

Additional comments:

1. References need to be corrected (for e.g. the first reference is not correctly cited). References after 36 are not listed. And the scope for improvement and list of incomplete references goes on.

We appreciate this comment. The references were verified and corrected.

REVIEWER: 2

Dr. Febisola I. Ajudua, Nelson Mandela Metropolitan University, Walter Sisulu University

The study highlights the gap in provision of social support services to patients diagnosed with TB in Mozambique, a high TB burden country.

1. The introduction adequately outlines the social and scientific value for this study. Although the diagram in page 21 outlines the contextual framework for the provision of support services, it is not titled. It is unclear whether this is what obtains currently, or what the author proposes for the provision of social support services to TB patients in need.

Thank you for your comment. The diagram on page 21 outlines the current provision of social support services in Mozambique. We specified it in the title of the figure and on the text regarding the "Structure of healthcare and social protection system", under the Methods section.

2. The author identified policy makers in areas identified to have the highest incidence of TB disease and describes how this is done. The author also demonstrates an understanding of the study method and clearly states this was a series of semi-structured interviews of selected policymakers. The limitation to the study method is described.

Thank you for your comment.

3. The findings are outlined clearly, and the author uses adequate statistical instruments in the review

of quantitative data. The discussion draws adequate comparison with international studies. It also addresses the main findings highlighting the effect of inadequate social support on health seeking behaviour, how the lack of social support could delay linkage to care and the need for better alignment of goals between the health and social support sectors. Although mentioned, it is not clear how policy makers in social support services can have a better understanding of the importance of social support for TB patients.

Thank you for your comment. Social support policy makers reported lacking clinical knowledge and stressed the need for allocation of staff with this competence such as health technicians and psychologists. On the other hand, improvement in communications between two sectors could improve their understanding of need of support for these people. This has been described under Theme 2 of the results section as follows: “Participants from social sector stressed that the sector, understandably, lacks clinical knowledge or competence. Participants reported that a health technician or a psychologist allocated to their offices would improve the identification of sick beneficiaries at the community level and promptly refer them to the healthcare sector for further management as appropriate”. Please, see lines 6-10 on page number 12.

4. The writing is clear and articulates the important aspects of reporting for a qualitative study. However, a few areas of the writing require attention. In the abstract, page 3, Line 16,17 – The authors need to specify semi-structured or in-depth interviews.

This has been changed under abstract section as follows: “Qualitative study with primary data collection through one-to-one in-depth interviews”. Please, see line 10 on page number 2.

5. The references are relevant to the subject, but a number of references are inaccurate in presentation provided on the list. Please pay particular attention to the listed references 1, 21, 25 and 35.

We appreciate this comment. The references were verified and corrected.

6. Diagram in page 21 is not titled, what area in the writing is it relevant for?

We submitted the manuscript according to the journal’s guidelines, which states that the legends of figures should come in a list after the references, separate from the figure. Figures were named figure 1 and 2 to match the legend on the provided list and text. We renamed the figure to enable easy match with the corresponding legend.

Hoping that this review will satisfy the readers, we would like to thank your consideration of our manuscript.

Sincerely,

Pedroso Nhassengo
(The corresponding Author)

VERSION 2 – REVIEW

REVIEWER	Rupani, Mihir Government Medical College Bhavnagar, Community Medicine
REVIEW RETURNED	26-Jul-2023
GENERAL COMMENTS	Congratulations for writing an important manuscript. All comments have been addressed. Cash transfer schemes have been demonstrated to be successful in high TB burden settings. The manuscript is an excellent opportunity to inform policy makers to

	implement the same even in low-resource settings. More of the same should be included in the discussion section, as per my opinion.
--	---